# Designing optimal COVID-19 testing stations locally: A discrete event simulation model applied on a university campus

**Michael Saidani** ⬥*, **Harrison Kim, Jinju Kim** ⬥

Department of Industrial and Enterprise Systems Engineering, Enterprise Systems Optimization Laboratory, University of Illinois at Urbana-Champaign, Urbana, Illinois, United States of America

* msaidani@illinois.edu

**Data Availability Statement:** All relevant data are within the manuscript and its S1 File, S1 Appendix, S1 and S2 Figs.

## Abstract

Providing sufficient testing capacities and accurate results in a time-efficient way are essential to prevent the spread and lower the curve of a health crisis, such as the COVID-19 pandemic. In line with recent research investigating how simulation-based models and tools could contribute to mitigating the impact of COVID-19, a discrete event simulation model is developed to design optimal saliva-based COVID-19 testing stations performing sensitive, non-invasive, and rapid-result RT-qPCR tests processing. This model aims to determine the adequate number of machines and operators required, as well as their allocation at different workstations, according to the resources available and the rate of samples to be tested per day. The model has been built and experienced using actual data and processes implemented on-campus at the University of Illinois at Urbana-Champaign, where an average of around 10,000 samples needed to be processed on a daily basis, representing at the end of August 2020 more than 2% of all the COVID-19 tests performed per day in the USA. It helped identify specific bottlenecks and associated areas of improvement in the process to save human resources and time. Practically, the overall approach, including the proposed modular discrete event simulation model, can easily be reused or modified to fit other contexts where local COVID-19 testing stations have to be implemented or optimized. It could notably support on-site managers and decision-makers in dimensioning testing stations by allocating the appropriate type and quantity of resources.

## Introduction

### Context and motivations

In accordance with the Centers for Disease Control and Prevention (CDC), proactive testing for COVID-19 infection is a key factor in determining where and how the SARS-CoV-2 virus is spreading within a population. The early identification of infected people leads to more rapid treatment and isolation for them, as well as for those who were exposed to them [1–3]. This type of monitoring is essential to reduce the spread of the disease (CDC, 2020). Fast and innovative solutions are indeed necessary to mitigate the consequences of the COVID-19 crisis [4]. In this line, since August 2020, the University of Illinois at Urbana-Champaign (UIUC) is

**Funding:** The authors received no specific funding for this work.

**Competing interests:** The authors have declared that no competing interests exist.

**Fig 1. COVID-19 testing workflow at the University of Illinois at Urbana-Champaign (UIUC).**

providing free COVID-19 diagnostic walk-in testing stations on campus. UIUC has actually implemented a time-efficient saliva-based COVID-19 test under an approved FDA Emergency Use Authorization [5].

This innovative saliva-based, specific nucleic acid (i.e., PCR), and rapid-result RT-qPCR COVID-19 testing process has been developed by the "COVID-19 SHIELD: Target, Test, Tell" team of UIUC [6]. According to the SHIELD team, direct saliva testing can address bottlenecks of time, cost, and supplies, enabling fast and frequent testings on a large scale. The participation throughout the semester by all students, faculty, and staff members is vital for collecting data to support the ongoing monitoring and tracking of the pandemic[7]. While the saliva-based process for COVID-19 testing, enables high-throughput, rapid, and scalable testing of a large population [8], the optimal design and dimensioning of the laboratory processing the samples are key to ensuring fast feedback to the people being tested.

On the one hand, UIUC has one of the most innovative on-campus COVID-19 testing programs in the United States of America (USA), offering up to 17 sites across campus [9]. On the other hand, it is of utmost importance to design an adequate on-site laboratory infrastructure and process–i.e., with the appropriate number of operators, machines, and adequate allocation of these resources–to test the samples in a time-efficient manner, as illustrated in Fig 1. The laboratory processing the saliva samples has achieved the regulatory compliance necessary to perform high-complexity testing under federal Clinical Laboratory Improvement Amendments guidelines. By the first day of classes (for the academic year 2020–2021), the goal was not only to administer more than 10,000 tests per day, but also to test all the samples collected and provide results within 24 hours. Such comprehensive testings allow for quick quarantine, public health contact tracing, and rapid delivery of any necessary medical care [10].

To put that in a national context, this number of 10,000 tests per day represented in August 2020 around two percent of all COVID-19 tests performed in the USA daily [10, 11]. The UIUC mass testing program and its associated platform have been touted as a model system, and have attracted the interest of many institutions [12]. On this basis, the overarching objective of this study is to figure out the optimal allocation of resources (i.e., operators, machines, and their allocation) to test and process a significant amount of samples locally on campus, by developing a simulation model that can be replicated and deployed in other contexts.

## Related work

The flexibility and adaptability of mobile health stations make them as commendable solutions to respond to pandemics, such as the COVID-19 crisis [13]. While they represent an untapped resource for healthcare systems, such mobile stations are still not widely implemented. An extensive literature review, through the evaluation of more than 50 articles, on the strengths

and weaknesses of mobile health stations in the United States, has been conducted recently [14]. A growing body of evidence shows that mobile health stations are particularly successful in delivering services directly at the curbside of communities in need. Yet, further work is necessary to augment the availability of mobile health care delivery [15]. Furthermore, in the context of the COVID-19 pandemic, with an increasing number of persons to be tested and "where limited intensive care resources can be overwhelmed by a large number of cases requiring admission in a short space of time" [16], managing healthcare demand and capacity is even more challenging. In this line, mobile testing stations appear to be a suitable solution to face the increasing demand for on-site COVID-19 testing services for workers and students. For instance, a design and engineering company has started designing mobile COVID-19 testing laboratories in conjunction with a laboratory equipment supplier to perform quick testings on large corporate and academic campuses [17]. The "mobile biosafety labs" developed can be deployed rapidly to locations that require COVID-19 testing for active or suspected cases of COVID-19. Their newly developed mobile lab can accommodate up to nine staff members, and two diagnostic machines (the first one for sample collection, the other one for testing) capable of testing 80 samples at a time, with a potential output of over 1,100 tests per day. In the present case, to reach the objective of 10,000 samples tested per day, ten mobile labs would be required, with 90 staff members and 20 machines, which represents a resource-intensive solution.

With this background, it becomes thus of utmost importance to take advantage of the capabilities offered by modeling and simulation tools to optimize the design and implementation of local and *ad hoc* COVID-19 testing stations. In this line, the following paragraphs focus on the applications of computer simulation to improve the performance of health services during the COVID-19 crisis. Lamé and Simmons (2020) discussed how simulation could be used in research works aiming at improving the quality, safety, and efficiency of healthcare systems [18]. In this line, Lamé and Dixon-Woods (2020) emphasized the substantial potential of simulation in healthcare systems, stating that "simulation can offer researchers access to events that can otherwise not be directly observed, and in a safe and controlled environment" and that "it is a flexible and pluripotent technique that can be used in multiple study designs in healthcare improvement research" [19]. Simulation notably allows many "what if?" scenarios to be tested in an efficient way for decision-making [20]. For instance, simulation-based failure mode analysis can be useful to identify the risks related to the readiness of the healthcare workers and emergency departments for the COVID-19 [21].

The use of realistic system models can actually help manage and mitigate a systemic crisis such as the COVID-19 pandemic [4]. Recently, researchers have discussed the role of systemic models to support better and agile management of the COVID-19 crisis, and suggested a structure for a COVID-19 decision-aid system based on three hierarchical layers [4]: (i) a top-level strategic level to master the crisis at a global level in a consistent fashion, (ii) an intermediate operational layer for operational decisions, based on the information captured from (iii) the tactical layer on a more local geographic scope. Typical examples of information and decisions at a tactical layer include the monitoring of equipment, beds, and ventilators used by COVID-19 patients in a given location. In a complementary way, Currie et al. (2020) recently started to investigate how simulation models can help reduce the impact of COVID-19 [22], as simulation models can be deployed for a variety of purposes, such as in the design of systems [23]. Currie et al. (2020) notably identified challenges resulting from the COVID-19 pandemic and discussed how simulation models could support decision-makers in making the most informed decisions [22]. The authors provided a mapping of the leading modeling techniques–namely system dynamics (SD), agent-based modeling (ABM), discrete event simulation (DES), and hybrid–on four scales (global, country or regional, organizational, individual), for three

emergency management phases (preparedness, response, recovery), and eleven COVID-related decisions, namely: quarantine, social distancing, end of lockdown, delivery of testing, targeting vaccination, hospital capacity, staffing, resource management, admission and discharge thresholds, other patients, health and wellbeing [22]. According to their mapping, for the delivery of testing (scope of the present paper), all three modeling techniques could be relevant. More specifically, for the modeling, simulation, and improvement of the COVID-19 testing process here, we argue that DES appears to be the most commendable approach to use.

In fact, DES is a method for simulating the behavior and performance of a real-life process, facility, or system [18]. In comparison with the principal features of SD and ABS models [24], DES models focus on processes that involve the use of a queue. By simulating the operation of a real-world system or process over time, DES models provide decision-makers with an evidence-based tool to develop and test operational solutions before implementation [25, 26]. DES models are also convenient to deploy at an operational and tactical level [27]. In addition, DES modeling includes three advantages that are commendable in the present case: (i) easy for the user to understand with the help of animations and graphics (available in the freely accessible AnyLogic PLE software package used in this study); (ii) flexibility to determine the behavior of entities; and (iii) modeling phase straightforward once the problem is clearly defined [4]. DES modeling is actually increasingly deployed in healthcare for improvement of services [27, 28], as an "effective decision-making tool for the optimal allocation of scarce health care resources to improve patient flow, while minimizing health care delivery costs and increasing patient satisfaction" [29]. Through a systematic literature review on the application of DES in healthcare [30], including more than 200 original research articles, it has been found that the applications of DES can be divided into four major classes: health and care systems operation, disease progression modeling, screening modeling, and health behavior modeling. For instance, DES can be deployed to determine the effectiveness of increasing the number of post-surgical inpatient beds on the proportion of patients admitted to a healthcare center [31]. Lamé et al. (2016) also applied DES to identify the sources of patient waiting times in an outpatient oncology clinic and to define relevant corrective actions [32]. By using DES to evaluate different scenarios, they quantitatively demonstrated that advanced preparation has the strongest potential for improving patient waiting times [32]. In addition, Rusnock et al. (2017) used DES to quantitatively model the mental workload of healthcare staff in an inpatient unit at a medical center [33]. The model was deployed to find the optimal idle time, average workload, and overload time of healthcare staff under different patient loads.

More recently, a stochastic DES model–freely available–has been developed to represent the critical dynamics of the intensive care admissions process for COVID-19 patients [16].It has been applied in large hospital in England for which the effect of several possible interventions were simulated. Particularly, model inputs were aligned with the action levers available to the planners, including duration of time at maximum capacity in order to inform workforce requirements. Almagor and Picascia (2020) evaluated the effectiveness of a COVID-19 contact tracing application using an agent-based model [34]. Fiore et al. (2021) deployed multi-agent simulations to estimate the daily testing capacity required to find and isolate a number of infected agents sufficient to break the transmission chain of COVID-19 infections [1]. Ghaffarzadegan (2021) developed a simulation model for what-if analyses to further monitor and mitigate the spread of COVID-19 in universities [35, 36]. In the present study, the objective is to build and deploy a new DES model to design optimal (in terms of time and resources) COVID-19 testing stations locally. The present research demonstrates how a high volume of saliva samples for COVID-19 testing can be achieved in a time-efficient way with proper process optimization under resource constraints and optimal allocation of testing machines and operators.

## Materials and methods

The need for quick and reliable COVID-19 testing has become crucial as students return to campus and employees to their workplace [37]. Both time and space are in limited supply for most of these places, which means building a novel and large structure for COVID-19 testing is not a convenient or practical solution [17]. While there was no pre-existing model available to process a significant number of COVID-19 tests on-campus on a daily basis, an initial process flow (table-based) has been proposed by the SHIELD team. The goal was to be capable of collecting and testing more than 10,000 samples in a given day, within a time window from 10 to 12 hours. In this paper, modeling and simulation tools are investigated and applied to verify and modify the process flow, and potentially draw newer and more effective process maps. Based on the model built and simulated for the University of Illinois detailed in this paper, a complementary objective of the present research work is to provide further insights and recommendations for decision-makers when designing and dimensioning testing stations in other contexts. In this line, the discrete event simulation (DES) model developed here can be replicated or scaled up for saliva testing station optimization in various situations, such as for testing in remote communities or concentrated cities.

To improve the scientific soundness and reproducibility of the DES model developed, the 20-item checklists aiming at "Strengthening The Reporting of Empirical Simulation Studies" (STRESS) [38] has been used, as reported in Table 1. The stepwise process for saliva sample testing is first mapped through a visual flowchart for better understanding, and a time-based Gantt chart is used to help visualize hotspots and potential areas of improvement, as illustrated in Fig 2. A DES model is then developed to run different scenarios (in terms of process configuration and resources allocation) to find the optimal testing process configuration to reach the

**Table 1. Application of the STRESS checklist [38] to the present DES model.**

| Category | Checklist item | Present simulation model |
|---|---|---|
| Objectives | Purpose of the model | Designing better COVID-19 testing stations |
| | Model outputs | Number of vials being processed on a daily basis |
| | Experimentation aims | Testing different configuration (in terms of operators and machines number and allocation) |
| Logic | Base model overview diagram | Gantt diagram of the testing process (S1 Appendix) |
| | Base model logic | Flowchart of the COVID-19 testing process (Fig 3) |
| | Scenario logic | Based on the hotspots (bottlenecks) identified |
| | Algorithms | Not applicable (N/A) |
| | Components | Number and allocation of operators and machines |
| Data | Data sources | The SHIELD team of the University of Illinois |
| | Input parameters | Time distribution and resources allocation (Table 2) |
| | Pre-processing | N/A |
| | Assumptions | Provided with the initial data by the experts from the SHIELD team (see values in Fig 3) |
| Experimentation | Initialisation | Initial configuration provided by the SHIELD team. See the initial transient regime for the first batch in Fig 7, before reaching the steady-state regime. |
| | Run-length | Two consecutive processing days (10 to 12 working hours per day) |
| | Estimation approach | Multiple replications (and box plots) for each scenario |
| Implementation | Software | AnyLogic PLE |
| | Random sampling | Triangular distribution function in AnyLogic (Monte Carlo simulation) |
| | Model execution | AnyLogic simulation engine FIFO (first in, first out) |
| | System specification | Intel Core i7-8550U, 1.80Ghz, 8.0GB RAM (Windows 10 Enterprise environment 64-bit) |
| Code access | Computer model | Supplementary digital file (DES_model.alp) |

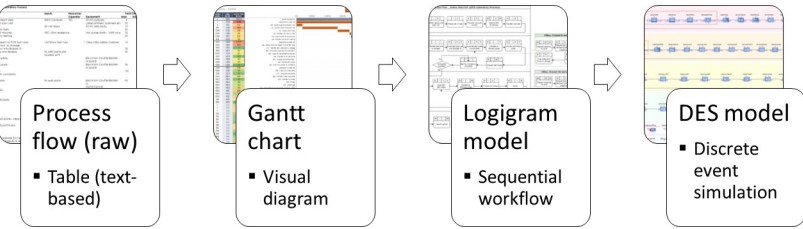

**Fig 2. Overview of the modeling approach.**

testing objective while minimizing the resources (operators and machines) deployed. To build and run the DES model, the software AnyLogic has been used as it is widely acknowledged for DES modeling [24, 39]. For instance, AnyLogic recently illustrated how such simulation-based models could help provide insight and decision support when applied to challenges like the COVID-19 outbreak [40]. In the present case, the AnyLogic Process Modeling Library, available in the free PLE version of AnyLogic, has been a helpful resource to model the real-world testing process in terms of agents (here, vials of saliva samples to be tested), processes (sequences of operations typically involving queues, delays, resources utilization), and resources (operators and machines), to optimize an existing mobile COVID-19 testing station and evaluate the impact of different configurations and resources allocations through simulations. In fact, the DES model is here a relevant stochastic tool for estimating probability distributions of potential outcomes by allowing for random variation in inputs over time [22]. DES models are typically deployed to model systems operation (e.g., a testing procedure) over time, where entities flow through several queues and activities. They are generally suitable for determining the impact of resource availability–operators and machines in the present case–on waiting times and the number of entities waiting in the queues or going through the system–vials to be tested here.

The present complete manuscript complements the initial study, and its associated two-page COVID-19 brief report, made by the present authors [41] to ensure rapid dissemination among the community in this context of the COVID-19 pandemic. In fact, after a synthetic literature survey providing background elements and inspiration sources, all the steps of the present research approach are now thoroughly detailed and illustrated to be clearly understandable, even by non-experts in modeling and simulation tools. Importantly, the DES model is made available (see S1 File) for researchers, managers, or decision-makers who want to reuse or adapt it in other contexts. Finally, the verification and validation of the DES model is a key point that is now further addressed in the discussion section, based on Sargent's recommendations [23], by comparing the outputs of the DES model with the data from the real situation on campus.

## Results

### Modeling phase

**Description and visualization of the COVID-19 saliva-based testing process.** The baseline or background information for this research work was the table-based process flow, given by the UIUC SHIELD team, including the innovative saliva-based process for large scale SARS-CoV-2 testing developed by a group of researchers at UIUC [8]. This model, as illustrated in Fig 2, allowed an initial understanding of the testing process, including a description of the different tasks to be performed, the resources required (operators and machines), and the time duration of each task.

Following the stepwise modeling approach depicted in Fig 2, the next step consisted of translating the table-based process flow into a Gantt diagram to better visualize the testing process and identify the time-consuming tasks. As this step is not mandatory for building the DES model, but could provide additional insights for decision-makers (e.g., a more visual understanding of the process on a timeline), two Gantt diagrams have been drawn and are available in S1 Appendix: one Gantt diagram with all tasks performed in serial in the minimum configuration (i.e., only one operator and one machine of each type available), and one Gantt diagram with the first proposition of improvement (ten operators and two testing machines available) allowing some tasks to be performed in parallel). The bottlenecks and key areas of improvement are highlighted by a box surrounded by a black border, in S1 and S2 Figs in S1 Appendix. On this basis, it has been possible to quickly identify appropriate ways to enhance the testing process, such as the tasks that could be parallelized by increasing the number of operators (e.g., between the tasks ID100 and ID140).

In parallel, the process flow for testing saliva samples has been mapped out using a logigram representation, in Fig 3, as a useful basis to build the DES model (see sub-section 3.2). Note that in Fig 3, the first logo–a vial–indicates the number of samples handled in each step; the

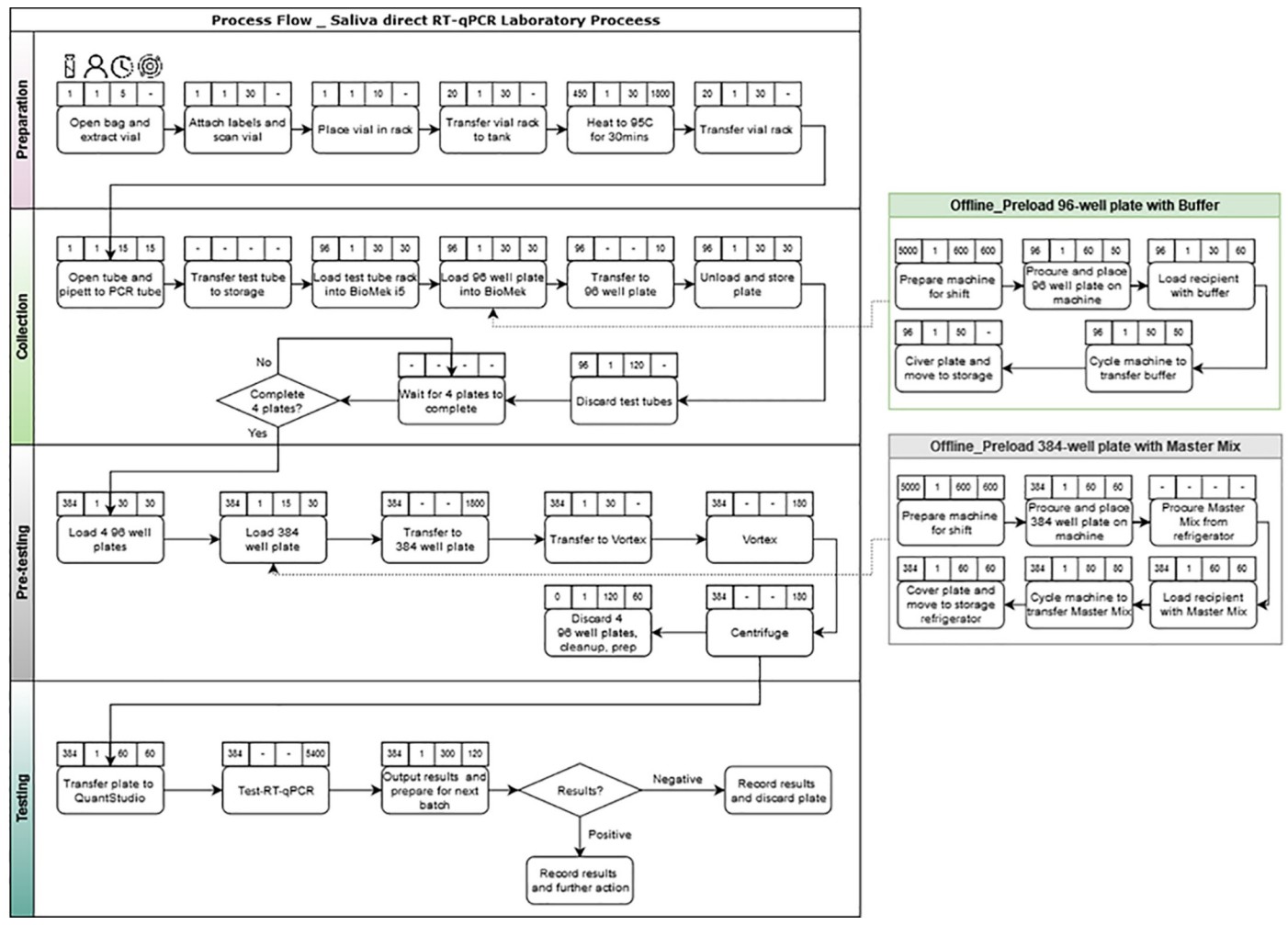

**Fig 3. Workflow model of the COVID-19 testing process, used for developing the DES model.**

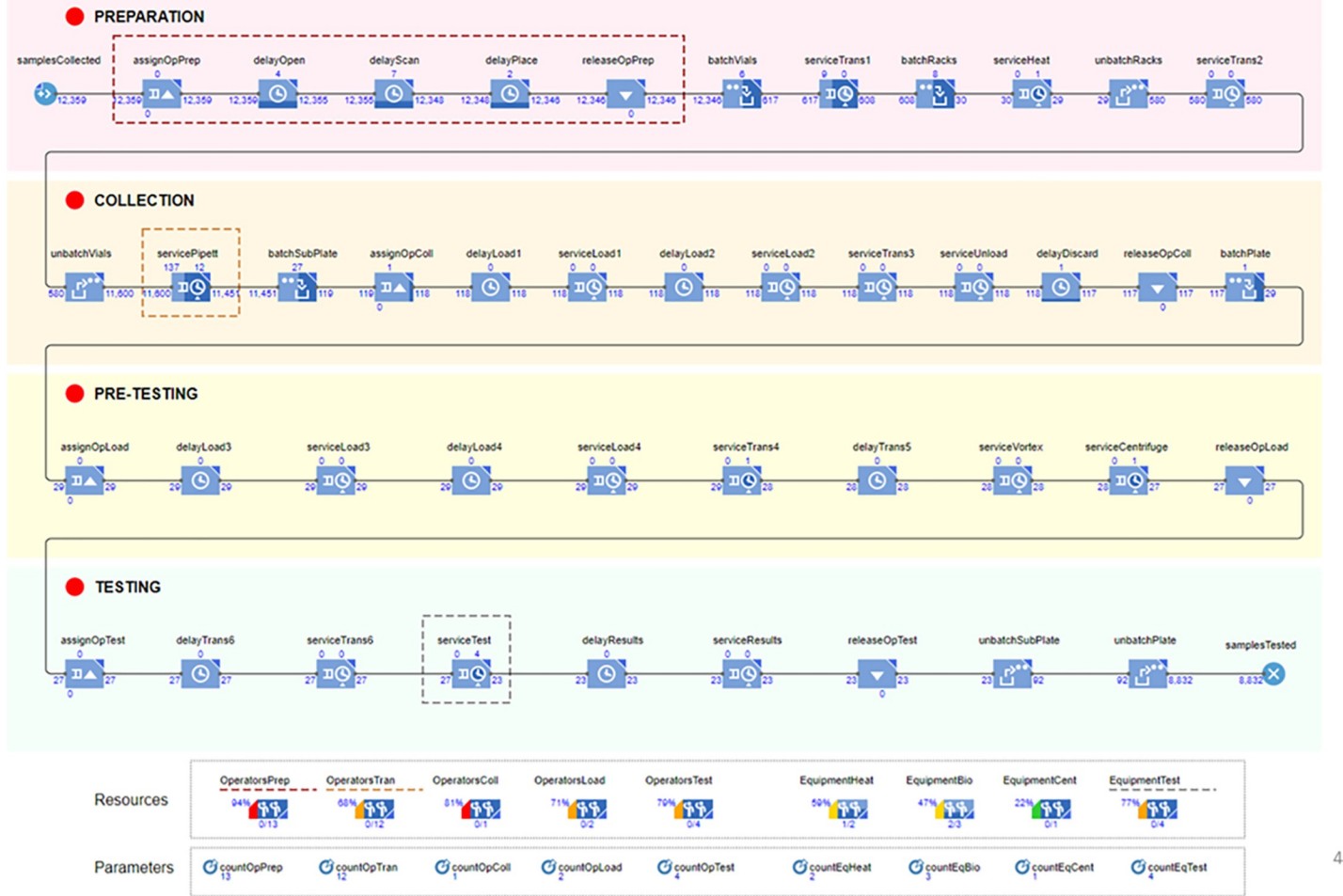

**Fig 4. Developed DES model, with pools of resources and parameters to optimize.**

second logo–a person–indicates the number of operators required to perform each task and process the associated number of vials; the third logo–a clock–indicates the time needed by the operator to perform and complete this task; and, the fourth logo–a gear–indicates whether or not a machine is required for a given task, and if any, the working time of the machine to process the associated number to vials for this task.

In the following sub-section, a discrete event simulation model is built and deployed to optimize the four main steps of the process flow, represented in Fig 3, in terms of resource allocation on-site. Note that the "offline" operations in Fig 3 refer to the operations that can be prepared in advance. The "offline" operations are out of scope for the present study, as it is assumed that a sufficient quantity of ready-to-be-used racks (including 96 and 384 well plates) is available to receive and carry the vials that have to be tested.

**Discrete event simulation (DES) model.** Fig 4 provides a complete overview of the DES model of the COVID-19 testing process for saliva samples deployed at UIUC, following the four main phases described in the previous sub-section, namely: preparation, collection, pre-testing, and testing. All key resources are modeled: a pool of operators and a pool of equipment, as illustrated in Fig 4. These resources are allocated to specific tasks, as listed in Table 2. Note that for some specific sequences of operations, the same operator is assigned (e.g., "assignOpPrep") to perform the whole sequence before being released (e.g., "releaseOpPrep")

**Table 2. Time distribution and resources allocation in the DES process.**

| Task | Time distribution (seconds) | Resources | In DES model |
|---|---|---|---|
| Open bag and extract vial | triangular(4,5,6) | Operator | OpPrep |
| Attach labels and scan vial | triangular(24,30,36) | Operator | OpPrep |
| Place vial in rack | triangular(8,10,12) | Operator | OpPrep |
| Transfer vial rack to tank | triangular(24,30,36) | Operator | OpTran |
| Heat to 95˚C | triangular(1800,1830,1860) | Machine | EqHeat |
| Transfer vial rack | triangular(24,30,36) | Operator | OpTran |
| Open tube and pipett to PCR tube | triangular(24,30,36) | Operator | OpTran |
| Load test tube rack into Biomek | triangular(24,30,36) | Operator | OpColl |
|  | constant(30) | Machine | EqBio |
| Load 96 well plate into Biomek | triangular(24,30,36) | Operator | OpColl |
|  | constant(30) | Machine | EqBio |
| Transfer to 96 well plate | constant(10) | Machine | EqBio |
| Unload and store plate | triangular(24,30,36) | Operator | OpColl |
|  | constant(30) | Machine | EqBio |
| Discard test tube | triangular(96,120,144) | Operator | OpColl |
| Load 4 96 well plate | triangular(24,30,36) | Operator | OpLoad |
|  | constant(30) | Machine | EqBio |
| Load 384 well plate | triangular(12,15,18) | Operator | OpLoad |
|  | constant(30) | Machine | EqBio |
| Transfer to 384 well plate | constant(1800) | Machine | EqBio |
| Transfer to Vortex | triangular(24,30,36) | Operator | OpLoad |
| Vortex | constant(180) | Machine | EqCent |
| Centrifuge | constant(180) | Machine | EqCent |
| Transfer plate to QuantiStudio | triangular(48,60,72) | Operator | OpTest |
|  | constant(60) | Machine | EqTest |
| Test-RT-qPCR | constant(5400) | Machine | EqTest |
| Output results and prepare for next batch | triangular(240,300,360) | Operator | OpTes |
|  | constant(120) | Machine | EqTest |

to handle a new (set of) vial(s). The time distribution of each task is given in Table 2, based on information provided by the "COVID-19 SHIELD: Target, Test, Tell" team. Note that while a constant time is used for equipment, a triangular distribution has been chosen, based on experts' knowledge, to model the variability of performance among the operators (with a distribution of +/- 20 percent around the mean value).

## Simulation phase and interpretation of the results

The DES model, developed using AnyLogic and based on the conceptual model described in the previous sub-section, has been run ten times for each of the different configurations in order to find out the optimal one, i.e., minimizing the number of resources used while achieving the time objective of 10,000 samples being tested in one working day (time window of 10–12 hours). In the present model, one replication of the DES model corresponds to one working day, to be coherent with the actual on-campus testing process, starting between 6 and 8 a.m. and finishing between 6 and 8 p.m. depending on the day and workload. The different configurations have been tested following an experiment plan, for which an extract is provided in the table at the bottom of Fig 5. Note that running each scenario more times does not change the box plots' features. This can be explained by the fact that the operator times

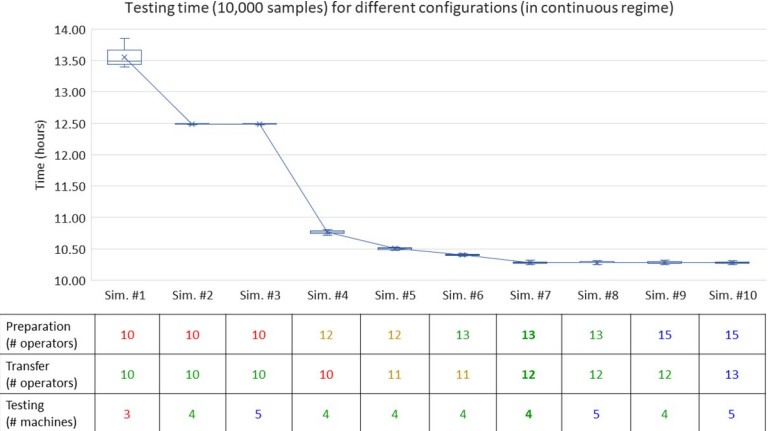

**Fig 5. Evaluation of scenarios through simulation runs (in established and continuous regime).**

(subjected to triangular distribution) are on average ten times lower than the machining times (constant time) (see Table 2). As such, in the present case, running the scenarios ten times allows being both time and cost-efficient, while generating sound simulation results.

After running a couple of simulations with realistic numbers for the set of parameters (i.e., varying the number of operators for each cluster of tasks, and the number of machines available), three key hotspots have been identified on the process flow, as highlighted through the dotted frames in Fig 4. These hotspots correspond to bottlenecks, where an accumulation (queue) of vials to be tested occurs, leading to slowing down the overall testing process flow. The first bottleneck is noticed when the number of operators allocated to the preparation of the vials is insufficient to deal with the number of vials collected for testing. The second one is also related to the number of operators allocated to the task "opening and pipetting", which needs to be performed individually for each sample. The third one is due to the time required (one hour and a half) to complete the "Test-RT-qPCR" task for a batch of 384 vials. As illustrated at the bottom of Fig 4, when running a DES simulation, the resources that are under-used (idle units) or overused (high utilization percentage) can be readily detected.

Fig 5 presents the time distribution for testing 10,000 samples times under different scenarios with ten replications for each scenario. To eliminate the three bottlenecks slowing down the whole process flow, the scenarios are built along three dimensions: (i) the number of operators for vials preparation, (ii) the number of resources allocated to the transfer operation, and (iii) the number of machines available to test a batch of 384 vials. Results in Fig 5 clearly show that two measures have a significant impact on testing times: adding more operators for preparation to a certain extent, and having sufficient testing machines available, as further discussed in the next paragraph. Another interesting insight is that, overall, there is a low variability induced by the operators' performances. Of course, the more operators and machines there are, the more time-efficient the process will be. Yet, the resources have to be optimized not only based on cost constraints but also to limit the number of operators working together at the same workplace or station to further prevent the spread of the virus.

In the first simulation (Sim. #1) listed in Fig 5, the insufficient number of both operators for preparation and testing machines creates two critical bottlenecks on the testing process flow, leading to a mean time above 13.5 hours to test 10,000 samples. Adding an extra testing machine (Sim. #2) allows reducing the meantime by one hour. For this configuration (to handle 10,000 samples) a day, having more than four machines (Sim. #3) does not bring any

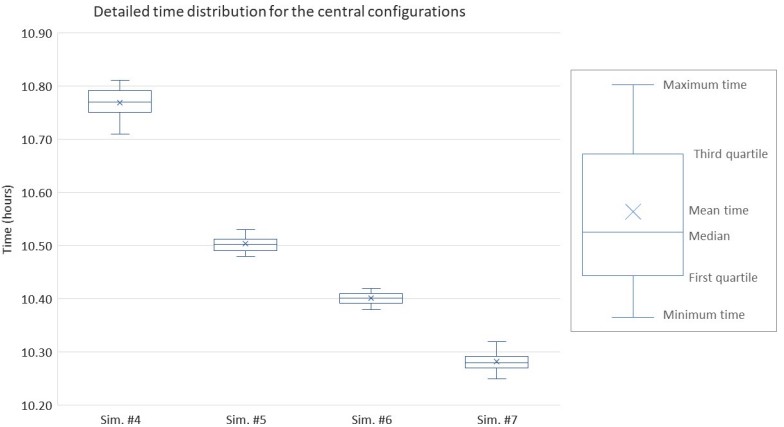

**Fig 6. Detailed box and whisker chart for key configurations (Sim. #4 to Sim. #7).**

improvement in terms of time efficiency. Augmenting the number of operators for preparation from 10 to 12 (Sim. #4) significantly decreases the queue, without completely solving this bottleneck. Also, as more vials are being treated simultaneously at the beginning of the testing process flow, this creates a queue for transfer operations (Sim. #4 to Sim. #6). In Sim. #7, no more bottlenecks are detected, and in this configuration, 10,000 saliva samples can be tested for COVID-19 in less than 10.5 hours, when operating in a continuous regime. Augmenting further the number of resources available (Sim. #8 to Sim. #10) does not significantly decrease the meantime.

In Fig 6, more details are given for the most promising configurations, i.e., the ones minimizing the use of resources while having a mean time below 11 hours. The boxplot shows the minimum, first quartile, median, third quartile, and maximum time after running ten simulations for each of these configurations. While the middle line of the box represents the median, dividing the time set into a bottom half and a top half, the "X" in the box represents the mean value. In all, as illustrated through the optimal steady state of Fig 7, the optimal resource allocation to test 10,000 samples within the time window available has been found (see Sim. #7 of Fig 5 and Table 3). Note that the transient regime only happens when a new testing center (re-) opens (e.g., after a break or holiday on campus). Other than that, it can be assumed that the

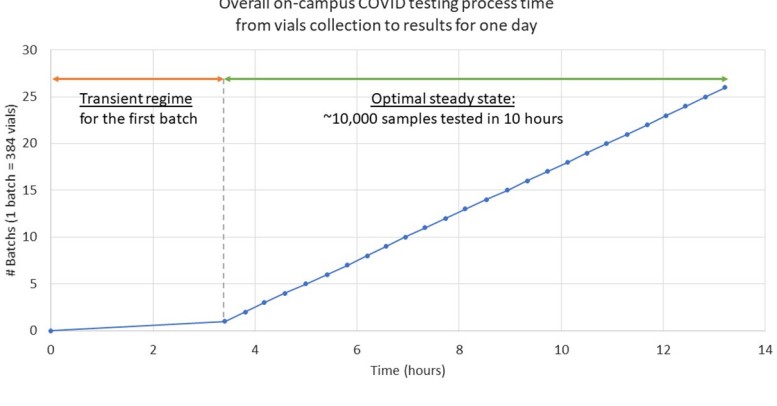

**Fig 7. Testing time of vials batches in transient and continuous operation.**

**Table 3. Optimal resources allocation, as a function of the number of samples to be tested.**

| Samples | OpPrep | OpTran | OpColl | OpLoad | OpTest | EqHeat | EqBio | EqCent | EqTest |
|---------|--------|--------|--------|--------|--------|--------|-------|--------|--------|
| 2000 | 3 | 2 | 1 | 1 | 2 | 1 | 1 | 1 | 2 |
| 4000 | 6 | 4 | 1 | 1 | 2 | 1 | 1 | 1 | 2 |
| 6000 | 9 | 7 | 1 | 1 | 3 | 1 | 2 | 1 | 3 |
| 8000 | 11 | 10 | 1 | 2 | 3 | 2 | 2 | 1 | 3 |
| 10000 | 13 | 12 | 1 | 2 | 4 | 2 | 3 | 1 | 4 |
| 12000 | 15 | 12 | 1 | 2 | 5 | 2 | 3 | 1 | 5 |
| 14000 | 19 | 13 | 2 | 3 | 6 | 2 | 3 | 1 | 6 |
| 16000 | 21 | 14 | 2 | 3 | 7 | 2 | 3 | 1 | 7 |
| 18000 | 23 | 16 | 2 | 3 | 7 | 3 | 3 | 1 | 7 |
| 20000 | 26 | 19 | 3 | 3 | 8 | 3 | 3 | 1 | 8 |

process can operate in a continuous regime, as the testing/processing center is operating on a continuous basis (i.e., 7 days a week) on campus. For information, the left part of the plot in Fig 7 indicates the additional time to consider when the process needs to be re-initialized or started from scratch.

## Discussion and implications

This study presented a DES model to help streamline operations at a large COVID-19 testing station on a university campus in the US. It has been shown that testing centers could benefit from the use of simulation models to increase the time-efficiency of their process while avoiding any overutilization of resources. This is particularly crucial in the current COVID-19 context, where millions of people are getting tested [42], and practitioners have to build new, or adapt, existing testing centers while making rapid but well-dimensioned design decisions. Through a DES model, it has been demonstrated that with a process flow designed and optimized in terms of resource use and allocation, it is feasible to achieve the goal of collecting, transporting, and testing 10,000 samples on-site per day with a reasonable quantity of resources mobilized.

The verification, validity, and reproducibility of simulation models are of utmost importance to serve scientific, societal, and practical benefits, notably for the advancement and reuse of operational knowledge [38]. Sargent (2013) provided and discussed practical approaches for the verification and validation of simulation models [23]. A given simulation model can be considered as valid when the model is an accurate representation of the real-world system, and when its domain of applicability possesses a satisfactory range of accuracy consistent with the intended application of the model [23]. Here, we are comparing the inputs and outputs of the DES model with the actual process and the number of samples being processed on the field. The model has been designed based on the inputs given by an expert from the "COVID-19 SHIELD: Target, Test, Tell", who actually developed and implement the testing process on campus at the University of Illinois at Urbana-Champaign.

The University of Illinois Urbana-Champaign has since released a data dashboard that displays daily information about the University's on-campus COVID-19 testing program [43], available at https://go.illinois.edu/COVIDTestingData. This dashboard displays the number of tests performed on a given day as well as the positivity rate. As shown in Fig 8, an average of 10,118 daily tests has been monitored for the first two weeks of class on-campus (for the academic year 2020/2021), which is well-aligned with the purpose and objective of the DES model, showing that the simulation output is close to the actual system output, in accordance with

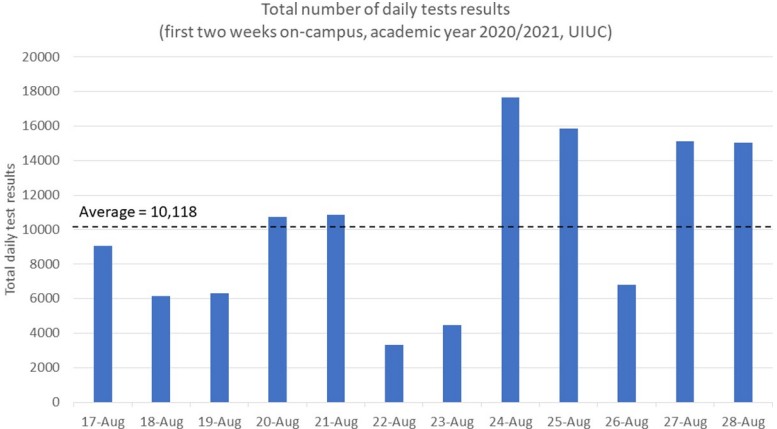

**Fig 8. Variability of the number of persons being tested on a daily basis (source: [43]).**

the event validity in [23]. On average, during the fall 2020 semester, UIUC conducted about 10,000 tests each weekday and about half that number during weekend days [44]. Also, in Fig 8, the higher number of tests performed is noticed at the beginning and end of each week, while it is on the weekend and on Wednesday that the lower number of persons being tested is recorded. As UIUC requires their students, staff, and faculty members to be tested twice a week to have access to on-campus facilities [7, 11], or at least once every four days, Monday/Thursday and Tuesday/Friday are the two couples of days with the higher number of tests to be processed.

On this basis, a timely and relevant line for future research would be to forecast the number of tests performed on a daily basis, in order to adapt the resources needed, e.g., to anticipate the days when a higher demand in terms of resources is required to ensure sufficient capacities and to provide testing results in a time-efficient manner (i.e., to reduce the time it takes to return COVID-19 test results). Real-time data from the UIUC testing program is expected to detect any emerging trends rapidly and to act quickly in response. While more data points are needed to build a sound prediction model (the testing policy on-campus is still evolving, and the trend cannot accurately be set until the new policy stabilizes), the DES model has been run to estimate the optimal resource allocation according to the number of samples to be tested per day.

The DES model developed in this paper can be launched quickly for scenario exploration to help adjust and refine the operations of a given testing program. In fact, it can be used to conduct what-if analyses. In practice, it can also be easily modified, for example, if some parts of the process flow evolved or if a new machine (e.g., a heating machine having a lower working time) can be implemented on the testing center. The optimal resources allocation (i.e., the minimal number of operators and machines for each task) depending on the number of saliva samples (from 2,000 to 20,000) to be tested in a single day (time window of 10 working hours available to process the test in a continuous regime) is given in Table 3 for the current process flow. As the statewide program, SHIELD Illinois, is currently working to increase current testing capacity to serve institutions nationally and entities in Illinois that have expressed interest in the new technology [45, 46], such results can be useful for decision-makers willing to implement a similar testing procedure in their respective contexts (e.g., an organization, a city) with more or fewer samples to be processed each day. As reminded by Lyng et al. (2021), the optimal use of COVID-19 tests will depend on different parameters such as the goals of testing, the population, or setting [2]. Last but not least, by following and applying the six principles of reporting simulation studies [38], the present DES model and its results can be reproduced, the model can be reused to investigate further hypotheses in the same application area or to

test the generalizability of this COVID-19 testing process in other situations. A promising line for future research would be to combine such simulation models with newly developed artificial intelligence techniques, e.g., automated machine learning [3], deep learning techniques [47] to further predicting and mitigating the COVID-19, as well as to share and maintain these data in a transparent and decentralized way using Blockchain technology [48].

## Supporting information

**S1 File. DES model.**
(ALP)

**S1 Appendix. Gantt diagrams of the testing process.**
(DOCX)

## Acknowledgments

This material is partially based upon the initial data and process flow provided by Christian Messmacher, from the SHIELD team. Any opinions, findings, and conclusions, or recommendations expressed in this publication are those of the authors. They do not necessarily reflect the views of the University of Illinois or the SHIELD Team.

## Author Contributions

**Conceptualization:** Michael Saidani, Harrison Kim.

**Data curation:** Michael Saidani, Jinju Kim.

**Formal analysis:** Michael Saidani.

**Investigation:** Michael Saidani.

**Methodology:** Michael Saidani, Jinju Kim.

**Project administration:** Harrison Kim.

**Resources:** Harrison Kim.

**Software:** Michael Saidani.

**Supervision:** Harrison Kim.

**Validation:** Michael Saidani, Harrison Kim.

**Visualization:** Michael Saidani, Jinju Kim.

**Writing – original draft:** Michael Saidani.

**Writing – review & editing:** Michael Saidani, Harrison Kim, Jinju Kim.

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
