## [Decision Letter · Decision Letter 0]

3 Jun 2021

PONE-D-21-16234

Designing optimal COVID-19 testing stations locally: a discrete event simulation model applied on a university campus

PLOS ONE

Dear Dr. Saidani,

Thank you for submitting your manuscript to PLOS ONE. After careful consideration, we feel that it has merit but does not fully meet PLOS ONE’s publication criteria as it currently stands. Therefore, we invite you to submit a revised version of the manuscript that addresses the points raised during the review process.

Based on the comments received from the reviewers and my own observation, I recommend major revisions for the article.

We look forward to receiving your revised manuscript.

Kind regards,

Thippa Reddy Gadekallu

Academic Editor

PLOS ONE

Journal Requirements:

2.Please amend either the title on the online submission form (via Edit Submission) or the title in the manuscript so that they are identical.

3.Thank you for stating the following in the Acknowledgments Section of your manuscript:

"This material is partially based upon the initial work and process flow provided by Christian

Messmacher, from the SHIELD team. Any opinions, findings, and conclusions, or recommendations

expressed in this publication are those of the authors and do not necessarily reflect the views of the

University of Illinois or the SHIELD Team."

"The authors received no specific funding for this work."

Additionally, because some of your funding information pertains to commercial funding, we ask you to provide an updated Competing Interests statement, declaring all sources of commercial funding.

In your Competing Interests statement, please confirm that your commercial funding does not alter your adherence to PLOS ONE Editorial policies and criteria by including the following statement: "This does not alter our adherence to PLOS ONE policies on sharing data and materials.” as detailed online in our guide for authors  http://journals.plos.org/plosone/s/competing-interests.  If this statement is not true and your adherence to PLOS policies on sharing data and materials is altered, please explain how.

Please include the updated Competing Interests Statement and Funding Statement in your cover letter. We will change the online submission form on your behalf.

Reviewers' comments:

Reviewer's Responses to Questions

**Comments to the Author**

1. Is the manuscript technically sound, and do the data support the conclusions?

Reviewer #1: Partly

Reviewer #2: Yes

2. Has the statistical analysis been performed appropriately and rigorously? 

Reviewer #1: No

Reviewer #2: Yes

3. Have the authors made all data underlying the findings in their manuscript fully available?

Reviewer #1: Yes

Reviewer #2: Yes

4. Is the manuscript presented in an intelligible fashion and written in standard English?

Reviewer #1: No

Reviewer #2: Yes

5. Review Comments to the Author

Reviewer #1: The authors have presented the research on. DESIGNING RESOURCE-EFFICIENT COVID-19 TESTING STATIONS LOCALLY:

A DISCRETE EVENT SIMULATION MODEL APPLIED ON A UNIVERSITY CAMPUS

1. The authors are advised to check the error in the title ( By mistake 0 is added in the title)

2. Secondly, overall, the language of the paper and grammatical error needs to be fixed throughout the paper, At present it is not upto the mark of the journal's expectations.

3. The font size and font style of the research paper needs to be according to the template instructions

4. Related work and literature review is very weak in the paper, authors are advised to be detailed in this section.

5. Authors have mentioned Fig 2. Overview of the modeling approach, but there is not figure 2 in the paper,

Lastly, authors are advised to format the references section and add below references

a) Srivastava D., Kohli R. and Gupta S. 2017 Advances in Computer and Computational Sciences (Springer) Implementation and statistical comparison of different edge detection techniques 211-228

b) Gomathi, S.; Kohli, R.; Soni, M.; Dhiman, G.; Nair, R. Pattern analysis: Predicting COVID-19 pandemic in India using AutoML. World J. Eng. 2020

Reviewer #2: • Introduction section can be extended to add the issues in the context of the existing work

• Literature review techniques have to be strengthened by including the issues in the current system and how the author proposes to overcome the same.

• There are some grammatical and editing problems in English. English presentation should be further polished

• The objective of the research should be clearly defined in the last paragraph of the introduction section.

• Add the advantages of the proposed system in one quoted line for justifying the proposed approach in the Introduction section. The authors can add the few advantages of deep learning for COVID -19 diagnosis. The following papers can be referred. Deep learning and medical image processing for coronavirus (COVID-19) pandemic: A survey.

Authors can refer An Incentive Based Approach for COVID-19 planning using Blockchain Technology.

6. PLOS authors have the option to publish the peer review history of their article (what does this mean?). If published, this will include your full peer review and any attached files.

Reviewer #1: No

Reviewer #2: No

---

## [Author Response · Author response to Decision Letter 0]

11 Jun 2021

Thank you for considering our research work to be published in PLOS ONE. We hope that the present responses and additional explanations, as well as the changes and add-ons made in the manuscript accordingly, address adequately the remarks and suggestions made by Reviewers #1 and #2.

The attached "Response to Reviewers" document includes specific responses - point-by-point - in blue font to each reviewer’s comments and suggestions. You will also find two versions of the revised manuscript: one with changes visible highlighted (showing clearly the changes made in response to the reviewers’ comments, as required), and one unmarked version.

---

## [Decision Letter · Decision Letter 1]

15 Jun 2021

Designing optimal COVID-19 testing stations locally: a discrete event simulation model applied on a university campus

PONE-D-21-16234R1

Dear Dr. Saidani,

We’re pleased to inform you that your manuscript has been judged scientifically suitable for publication and will be formally accepted for publication once it meets all outstanding technical requirements.

Kind regards,

Thippa Reddy Gadekallu

Academic Editor

PLOS ONE

Additional Editor Comments (optional):

Reviewers' comments:

Reviewer's Responses to Questions

**Comments to the Author**

1. If the authors have adequately addressed your comments raised in a previous round of review and you feel that this manuscript is now acceptable for publication, you may indicate that here to bypass the “Comments to the Author” section, enter your conflict of interest statement in the “Confidential to Editor” section, and submit your "Accept" recommendation.

Reviewer #1: All comments have been addressed

Reviewer #2: All comments have been addressed

2. Is the manuscript technically sound, and do the data support the conclusions?

Reviewer #1: Yes

Reviewer #2: Yes

3. Has the statistical analysis been performed appropriately and rigorously? 

Reviewer #1: Yes

Reviewer #2: Yes

4. Have the authors made all data underlying the findings in their manuscript fully available?

Reviewer #1: Yes

Reviewer #2: Yes

5. Is the manuscript presented in an intelligible fashion and written in standard English?

Reviewer #1: Yes

Reviewer #2: Yes

6. Review Comments to the Author

Reviewer #1: All comments have been addressed. However, it hits at strongly recommended to a th reference suggested previously

Reviewer #2: The authors have addressed all of my comments. The paper can can be accepted in the current format. Thank you

7. PLOS authors have the option to publish the peer review history of their article (what does this mean?). If published, this will include your full peer review and any attached files.

Reviewer #1: No

Reviewer #2: No

---

## [Editor Report · Acceptance letter]

18 Jun 2021

PONE-D-21-16234R1 

Designing optimal COVID-19 testing stations locally: a discrete event simulation model applied on a university campus 

Dear Dr. Saidani:

I'm pleased to inform you that your manuscript has been deemed suitable for publication in PLOS ONE. Congratulations! Your manuscript is now with our production department. 

Kind regards, 

on behalf of

Dr. Thippa Reddy Gadekallu 

Academic Editor

PLOS ONE